# Compact Generalized Non-local Network

**Kaiyu Yue**[†,§]   **Ming Sun**[†]   **Yuchen Yuan**[†]   **Feng Zhou**[‡]   **Errui Ding**[†]   **Fuxin Xu**[§]

[†]Baidu VIS   [‡]Baidu Research   [§]Central South University

{yuekaiyu, sunming05, yuanyuchen02, zhoufeng09, dingerrui}@baidu.com
fxxu@csu.edu.cn

## Abstract

The non-local module [27] is designed for capturing long-range spatio-temporal dependencies in images and videos. Although having shown excellent performance, it lacks the mechanism to model the interactions between positions across channels, which are of vital importance in recognizing fine-grained objects and actions. To address this limitation, we generalize the non-local module and take the correlations between the positions of any two channels into account. This extension utilizes the compact representation for multiple kernel functions with Taylor expansion that makes the generalized non-local module in a fast and low-complexity computation flow. Moreover, we implement our generalized non-local method within channel groups to ease the optimization. Experimental results illustrate the clear-cut improvements and practical applicability of the generalized non-local module on both fine-grained object recognition and video classification. Code is available at:
https://github.com/KaiyuYue/cgnl-network.pytorch.

## 1   Introduction

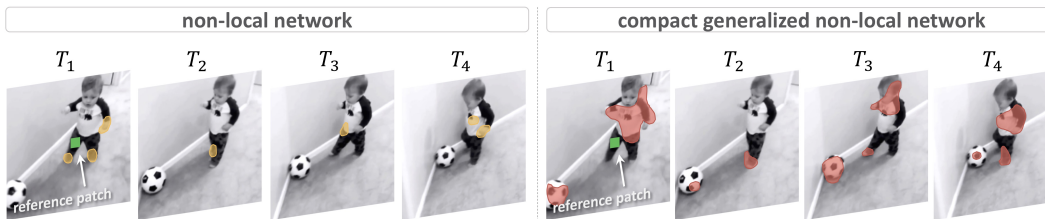

**Figure 1:** Comparison between non-local (NL) and **compact generalized non-local (CGNL)** networks on recognizing an action video of kicking the ball. Given the *reference patch* (green rectangle) in the first frame, we visualize for each method the highly related responses in the other frames by thresholding the feature space. CGNL network out-performs the original NL network in capturing the ball that is not only in long-range distance from the reference patch but also corresponds to different channels in the feature map.

Capturing spatio-temporal dependencies between spatial pixels or temporal frames plays a key role in the tasks of fine-grained object and action classification. Modeling such interactions among images and videos is the major topic of various feature extraction techniques, including SIFT, LBP, Dense Trajectory [26], etc. In the past few years, deep neural network automates the feature designing pipeline by stacking multiple end-to-end convolutional or recurrent modules, where each of them processes correlation within spatial or temporal local regions. In general, capturing the long-range dependencies among images or videos still requires multiple stacking of these modules, which greatly hinders the learning and inference efficiency. A recent work [16] also suggests that stacking more layers cannot always increase the effective receptive fields to capture enough local relations.

Inspired by the classical non-local means for image filtering, the recently proposed non-local neural network [27] addresses this challenge by directly modeling the correlation between any two positions in the feature maps in a single module. Without bells and whistles, the non-local method can greatly improve the performances of existing networks on many video classification benchmarks. Despite its great performances, the original non-local network only considers the global spatio-temporal correlation by merging channels, and it might miss the subtle but important cross-channel clues for discriminating fine-grained objects or actions. For instance, the body, the ball and their interaction are all necessary for describing the action of kicking the ball in Fig. 1, while the original non-local operation learns to focus on the body part relations but neglect the body-ball interactions that usually correspond to different channels of the input features.

To improve the effectiveness in fine-grained object and action recognition tasks, this work extends the non-local module by learning explicit correlations among all of the elements across the channels. First, this extension scale-ups the representation power of the non-local operation to attend the interaction between subtle object parts (*e.g.*, the body and ball in Fig. 1). Second, we propose its compact representation for various kernel functions to address the high computation burden issue. We show that as a self-contained module, the compact generalized non-local (CGNL) module provides steady improvements in classification tasks. Third, we also investigate the grouped CGNL blocks, which model the correlations across channels within each group.

We evaluate the proposed CGNL method on the task of fine-grained classification and action recognition. Extensive experimental results show that: 1) The CGNL network are easy to optimize as the original non-local network; 2) Compared with the non-local module, CGNL module enjoys capturing richer features and dense clues for prediction, as shown in Figure 1, which leads to results substantially better than those of the original non-local module. Moreover, in the appendix of extensional experiments, the CGNL network can also promise a higher accuracy than the baseline on the large-scale ImageNet dataset [20].

## 2 Related Works

**Channel Correlations:** The mechanism of sharing the same conv kernel among channels of a layer in a ConvNet [12] can be seen as a basic way to capture correlations among channels, which aggregates the channels of feature maps by the operation of sum pooling. The SENet [10] may be the first work that explicitly models the interdependencies between the channels of its spatial features. It aims to select the useful feature maps and suppress the others, and only considers the global information of each channel. Inspired by [27], we present the generalized non-local (GNL) module, which generalizes the non-local (NL) module to learn the correlations between any two positions across the channels. Compared to the SENet, we model the interdependencies among channels in an explicit and dense manner.

**Compact Representation:** After further investigation, we find that the non-local module contains a second-order feature space (Sect.3.1), which is used widely in previous computer vision tasks, *e.g.*, SIFT [15], Fisher encoding [17], Bilinear model [14] [5] and segmentation task [2]. However, such second-order feature space involves high dimensions and heavy computational burdens. In the area of kernel learning [21], there are many prior works such as compact bilinear pooling (CBP) [5] that uses the Tensor Sketching [18] to address this problem. But this type of method is not perfect yet. Because the it cannot produce a light computation to the various size of sketching vectors. Fortunately, in mathematics, the whole non-local operation can be viewed as a trilinear formation. It can be fast computed with the associative law of matrix production. To the other types of pairwise function, such as Embedded Gaussian or RBF [19], we propose a tight approximation for them by using the Taylor expansion.

## 3 Approach

In this section, we introduce a general formulation of the proposed general non-local operation. We then show that the original non-local and the bilinear pooling are special cases of this formulation. After that, we illustrate that the general non-local operation can be seen as a modality in the trilinear matrix production and show how to implement our generalized non-local (GNL) module in a compact representations.

## 3.1 Review of Non-local Operation

We begin by briefly reviewing the original non-local operation [27] in matrix form. Suppose that an image or video is given to the network and let $\mathbf{X} \in \mathbb{R}^{N \times C}$ denote (see notation[1]) the input feature map of the non-local module, where $C$ is the number of channels. For the sake of notation clarity, we collapse all the spatial (width $W$ and height $H$) and temporal (video length $T$) positions in one dimension, *i.e.*, $N = HW$ or $N = HWT$. To capture long-range dependencies across the whole feature map, the original non-local operation computes the response $\mathbf{Y} \in \mathbb{R}^{N \times C}$ as the weighted sum of the features at all positions,

$$\mathbf{Y} = f\big(\theta(\mathbf{X}), \phi(\mathbf{X})\big) g(\mathbf{X}), \tag{1}$$

where $\theta(\cdot), \phi(\cdot), g(\cdot)$ are learnable transformations on the input. In [27], the authors suggest using $1 \times 1$ or $1 \times 1 \times 1$ convolution for simplicity, *i.e.*, the transformations can be written as

$$\theta(\mathbf{X}) = \mathbf{X}\mathbf{W}_\theta \in \mathbb{R}^{N \times C}, \quad \phi(\mathbf{X}) = \mathbf{X}\mathbf{W}_\phi \in \mathbb{R}^{N \times C}, \quad g(\mathbf{X}) = \mathbf{X}\mathbf{W}_g \in \mathbb{R}^{N \times C}, \tag{2}$$

parameterized by the weight matrices $\mathbf{W}_\theta, \mathbf{W}_\phi, \mathbf{W}_g \in \mathbb{R}^{C \times C}$ respectively. The pairwise function $f(\cdot, \cdot) : \mathbb{R}^{N \times C} \times \mathbb{R}^{N \times C} \rightarrow \mathbb{R}^{N \times N}$ computes the affinity between all positions (space or space-time). There are multiple choices for $f$, among which dot-product is perhaps the simplest one, *i.e.*,

$$\mathbf{f}\big(\theta(\mathbf{X}), \phi(\mathbf{X})\big) = \theta(\mathbf{X})\phi(\mathbf{X})^\top. \tag{3}$$

Plugging Eq. 2 and Eq. 3 into Eq. 1 yields a trilinear interpretation of the non-local operation,

$$\mathbf{Y} = \mathbf{X}\mathbf{W}_\theta \mathbf{W}_\phi^\top \mathbf{X}^\top \mathbf{X}\mathbf{W}_g, \tag{4}$$

where the pairwise matrix $\mathbf{X}\mathbf{W}_\theta \mathbf{W}_\phi^\top \mathbf{X}^\top \in \mathbb{R}^{N \times N}$ encodes the similarity between any locations of the input feature. The effect of non-local operation can be related to the self-attention module [1] based on the fact that each position (row) in the result $\mathbf{Y}$ is a linear combination of all the positions (rows) of $\mathbf{X}\mathbf{W}_g$ weighted by the corresponding row of the pairwise matrix.

## 3.2 Review of Bilinear Pooling

Analogous to the conventional kernel trick [21], the idea of bilinear pooling [14] has recently been adopted in ConvNets for enhancing the feature representation in various tasks, such as fine-grained classification, person re-id, action recognition. At a glance, bilinear pooling models pairwise feature interactions using explicit outer product at the final classification layer:

$$\mathbf{Z} = \mathbf{X}^\top \mathbf{X} \in \mathbb{R}^{C \times C}, \tag{5}$$

where $\mathbf{X} \in \mathbb{R}^{N \times C}$ is the input feature map generated by the last convolutional layer. Each element of the final descriptor $z_{c_1 c_2} = \sum_n x_{nc_1} x_{nc_2}$ sum-pools at each location $n = 1, \cdots, N$ the bilinear product $x_{nc_1} x_{nc_2}$ of the corresponding channel pair $c_1, c_2 = 1, \cdots, C$.

Despite the distinct design motivation, it is interesting to see that bilinear pooling (Eq. 5) can be viewed as a special case of the second-order term (Eq. 3) in the non-local operation if we consider,

$$\theta(\mathbf{X}) = \mathbf{X}^\top \in \mathbb{R}^{C \times N}, \quad \phi(\mathbf{X}) = \mathbf{X}^\top \in \mathbb{R}^{C \times N}. \tag{6}$$

## 3.3 Generalized Non-local Operation

The original non-local operation aims to directly capture long-range dependencies between any two positions in one layer. However, such dependencies are encoded in a joint location-wise matrix $f(\theta(\mathbf{X}), \phi(\mathbf{X}))$ by aggregating all channel information together. On the other hand, channel-wise correlation has been recently explored in both discriminative [14] and generative [24] models through the covariance analysis across channels. Inspired by these works, we generalize the original non-local operation to model long-range dependencies between any positions of any channels.

We first reshape the output of the transformations (Eq. 2) on $\mathbf{X}$ by merging channel into position:

$$\theta(\mathbf{X}) = \text{vec}(\mathbf{X}\mathbf{W}_\theta) \in \mathbb{R}^{NC}, \phi(\mathbf{X}) = \text{vec}(\mathbf{X}\mathbf{W}_\phi) \in \mathbb{R}^{NC}, g(\mathbf{X}) = \text{vec}(\mathbf{X}\mathbf{W}_g) \in \mathbb{R}^{NC}. \quad (7)$$

By lifting the row space of the underlying transformations, our generalized non-local (GNL) operation pursues the same goal of Eq. 1 that computes the response $\mathbf{Y} \in \mathbb{R}^{N \times C}$ as:

$$\text{vec}(\mathbf{Y}) = f\big(\text{vec}(\mathbf{X}\mathbf{W}_\theta), \text{vec}(\mathbf{X}\mathbf{W}_\phi)\big) \text{vec}(\mathbf{X}\mathbf{W}_g). \quad (8)$$

Compared to the original non-local operation (Eq. 4), GNL utilizes a more general pairwise function $f(\cdot, \cdot) : \mathbb{R}^{NC} \times \mathbb{R}^{NC} \rightarrow \mathbb{R}^{NC \times NC}$ that can differentiate between pairs of same location but at different channels. This richer similarity greatly augments the non-local operation in discriminating fine-grained object parts or action snippets that usually correspond to channels of the input feature. Compared to the bilinear pooling (Eq. 5) that can only be used after the last convolutional layer, GNL maintains the input size and can thus be flexibly plugged between any network blocks. In addition, bilinear pooling neglects the spatial correlation which, however, is preserved in GNL.

Recently, the idea of dividing channels into groups has been established as a very effective technique in increasing the capacity of ConvNets. Well-known examples include Xception [3], MobileNet [9], ShuffleNet [31], ResNeXt [29] and Group Normalization [28]. Given its simplicity and independence, we also realize the channel grouping idea in GNL by grouping all $C$ channels into $G$ groups, each of which contains $C' = C/G$ channels of the input feature. We then perform GNL operation independently for each group to compute $\mathbf{Y}'$ and concatenate the results along the channel dimension to restore the full response $\mathbf{Y}$.

### 3.4 Compact Representation

A straightforward implementation of GNL (Eq. 8) is prohibitive as the quadratic increase with respect to the channel number $C$ in the presence of the $NC \times NC$ pairwise matrix. Although the channel grouping technique can reduce the channel number from $C$ to $C/G$, the overall computational complexity is still much higher than the original non-local operation. To mitigate this problem, this section proposes a compact representation that leads to an affordable approximation for GNL.

Let us denote $\boldsymbol{\theta} = \text{vec}(\mathbf{X}\mathbf{W}_\theta)$, $\boldsymbol{\phi} = \text{vec}(\mathbf{X}\mathbf{W}_\phi)$ and $\boldsymbol{g} = \text{vec}(\mathbf{X}\mathbf{W}_g)$, each of which is a $NC$-D vector column. Without loss of generality, we assume $f$ is a general kernel function (*e.g.*, RBF, bilinear, etc.) that computes a $NC \times NC$ matrix composed by the elements,

$$\big[f(\boldsymbol{\theta}, \boldsymbol{\phi})\big]_{ij} \approx \sum_{p=0}^{P} \alpha_p^2 (\theta_i \phi_j)^p, \quad (9)$$

which can be approximated by Taylor series up to certain order $P$. The coefficient $\alpha_p$ can be computed in closed form once the kernel function is known. Taking RBF kernel for example,

$$[f(\boldsymbol{\theta}, \boldsymbol{\phi})]_{ij} = \exp(-\gamma \|\theta_i - \phi_j\|^2) \approx \sum_{p=0}^{P} \beta \frac{(2\gamma)^p}{p!} (\theta_i \phi_j)^p, \quad (10)$$

where $\alpha_p^2 = \beta \frac{(2\gamma)^p}{p!}$ and $\beta = \exp\big(-\gamma(\|\boldsymbol{\theta}\|^2 + \|\boldsymbol{\phi}\|^2)\big)$ is a constant and $\beta = \exp(-2\gamma)$ if the input vectors $\boldsymbol{\theta}$ and $\boldsymbol{\phi}$ are $\ell2$-normalized. By introducing two matrices,

$$\boldsymbol{\Theta} = [\alpha_0 \boldsymbol{\theta}^0, \cdots, \alpha_P \boldsymbol{\theta}^P] \in \mathbb{R}^{NC \times (P+1)}, \quad \boldsymbol{\Phi} = [\alpha_0 \boldsymbol{\phi}^0, \cdots, \alpha_P \boldsymbol{\phi}^P] \in \mathbb{R}^{NC \times (P+1)} \quad (11)$$

our compact generalized non-local (CGNL) operation approximates Eq. 8 via a trilinear equation,

$$\text{vec}(\mathbf{Y}) \approx \boldsymbol{\Theta} \boldsymbol{\Phi}^\top \boldsymbol{g}. \quad (12)$$

At first glance, the above approximation still involves the computation of a large pairwise matrix $\boldsymbol{\Theta}\boldsymbol{\Phi}^\top \in \mathbb{R}^{NC \times NC}$. Fortunately, the order of Taylor series is usually relatively small $P \ll NC$. According to the associative law, we could alternatively compute the vector $\boldsymbol{z} = \boldsymbol{\Phi}^\top \boldsymbol{g} \in \mathbb{R}^{P+1}$ first and then calculate $\boldsymbol{\Theta}\boldsymbol{z}$ in a much smaller complexity of $\mathcal{O}(NC(P+1))$. In another view, the process that this bilinear form $\boldsymbol{\Phi}^\top \boldsymbol{g}$ is squeezed into scalars can be treated as a related concept of the SE module [10].

**Complexity analysis:** Table 1 compares the computational complexity of CGNL network with the GNL ones. We cannot afford for directly computing GNL operation because of its huge complexity of $\mathcal{O}(2(NC)^2)$ in both time and space. Instead, our compact method dramatically eases the heavy calculation to $\mathcal{O}(NC(P+1))$.

**Table 1:** Complexity comparison of GNL and CGNL operations, where $N$ and $C$ indicate the number of positions and channels respectively.

|  | General NL Method | CGNL Method |
|---|---|---|
| Strategy | $f(\mathbf{\Theta\Phi}^\top)\boldsymbol{g}$ | $\mathbf{\Theta\Phi}^\top\boldsymbol{g}$ |
| Time | $\mathcal{O}(2(NC)^2)$ | $\mathcal{O}(NC(P+1))$ |
| Space | $\mathcal{O}(2(NC)^2)$ | $\mathcal{O}(NC(P+1))$ |

## 3.5 Implementation Details

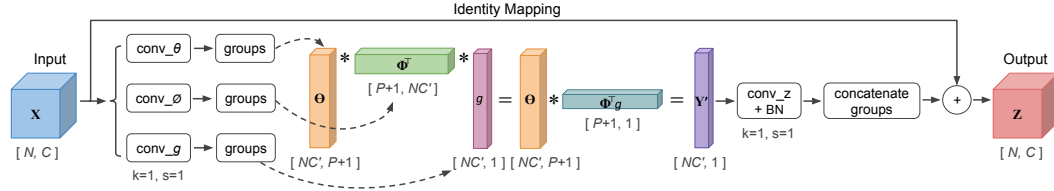

**Figure 2: Grouped compact generalized non-local (CGNL) module.** The feature maps are shown with the shape of their tensors, *e.g.*, $[C, N]$, where $N = THW$ or $N = HW$. The feature maps will be divided along channels into multiple groups after three conv layers whose kernel size and stride both equals 1 (k = 1, s = 1). The channels dimension is grouped into $C' = C/G$, where $G$ is a group number. The compact representations for generalized non-local module are build within each group. $P$ indicates the order of Taylor expansion for kernel functions.

Fig. 2 illustrates the workflow of how CGNL module processes a feature map $\mathbf{X}$ of the size $N \times C$, where $N = H \times W$ or $N = T \times H \times W$. $\mathbf{X}$ is first fed into three $1 \times 1 \times 1$ convolutional layers that are described by the weights $W_\theta, W_\phi, W_g$ respectively in Eq. 7. To improve the capacity of neural networks, the channel grouping idea [29, 28] is then applied to divide the transformed feature along the channel dimension into $G$ groups. As shown in Fig. 2, we approximate for each group the GNL operation (Eq. 8) using the Taylor series according to Eq. 12. To achieve generality and compatibility with existing neural network blocks, the CGNL block is implemented by wrapping Eq. 8 in an identity mapping of the input as in residual learning [8]:

$$\mathbf{Z} = concat(BN(\mathbf{Y}'\mathbf{W}_z)) + \mathbf{X}, \tag{13}$$

where $\mathbf{W}_z \in \mathbb{R}^{C \times C}$ denotes a $1 \times 1$ or $1 \times 1 \times 1$ convolution layer followed by a Batch Normalization [11] in each group.

# 4 Experiments

## 4.1 Datasets

We evaluate the CGNL network on multiple tasks, including fine-grained classification and action recognition. For fine-grained classification, we experiment on the Birds-200-2011 (CUB) dataset [25], which contains 11788 images of 200 bird categories. For action recognition, we experiment on two challenging datasets, Mini-Kinetics [30] and UCF101 [22]. The Mini-Kinetics dataset contains 200 action categories. Due to some video links are unavaliable to download, we use 78265 videos for training and 4986 videos for validation. The UCF101 dataset contains 101 actions, which are separated into 25 groups with 4-7 videos of each action in a group.

## 4.2 Baselines

Given the steady performance and efficiency, the ResNet [8] series (ResNet-50 and ResNet-101) are adopted as our baselines. For video tasks, we keep the same architecture configuration with [27], where the temporal dimension is trivially addressed by max pooling. Following [27] the convolutional layers in the baselines are implemented as $1 \times k \times k$ kernels, and we insert our CGNL blocks into

**Table 2: Ablations.** Top1 and top5 accuracy (%) on various datasets.

**(a)** Results of adding 1 CGNL block on CUB. The kernel of dot production achieves the best result. The accuracies of others are at the edge of baselines.

| model | top1 | top5 |
|---|---|---|
| R-50. | 84.05 | 96.00 |
| Dot Production | 85.14 | 96.88 |
| Gaussian RBF | 84.10 | 95.78 |
| Embedded Gaussian | 84.01 | 96.08 |

**(c)** Results of channel grouped CGNL networks on CUB. A few groups can boost the performance. But more groups tend to prevent the CGNL block from capturing the correlations between positions across channels.

| model | groups | top1 | top5 |
|---|---|---|---|
| R-101 | - | 85.05 | 96.70 |
| + 1 CGNL block | 1 | 86.17 | 97.82 |
| | 4 | 86.24 | 97.05 |
| | 8 | 86.35 | 97.86 |
| | 16 | 86.13 | 96.75 |
| | 32 | 86.04 | 96.69 |

| model | groups | top1 | top5 |
|---|---|---|---|
| R-101 | - | 85.05 | 96.70 |
| + 5 CGNL block | 1 | 86.01 | 95.97 |
| | 4 | 86.19 | 96.07 |
| | 8 | 86.24 | 97.23 |
| | 16 | 86.43 | 98.89 |
| | 32 | 86.10 | 97.13 |

**(b)** Results of comparison on UCF-101. Note that CGNL network is not grouped in channel.

| model | top1 | top5 |
|---|---|---|
| R-50. | 81.62 | 94.62 |
| + 1 NL block | 82.88 | 95.74 |
| + 1 CGNL block | 83.38 | 95.42 |

**(d)** Results of grouped CGNL networks on Mini-Kinetics. More groups help the CGNL networks improve top1 accuracy obveriously.

| model | gorups | top1 | top5 |
|---|---|---|---|
| R-50 | - | 75.54 | 92.16 |
| + 1 CGNL block | 1 | 77.16 | 93.56 |
| | 4 | 77.56 | 93.00 |
| | 8 | 77.76 | 93.18 |

| model | gorups | top1 | top5 |
|---|---|---|---|
| R-101 | - | 77.44 | 93.18 |
| + 1 CGNL block | 1 | 78.79 | 93.64 |
| | 4 | 79.06 | 93.54 |
| | 8 | 79.54 | 93.84 |

the network to turn them into compact generalized non-local (CGNL) networks. We investigate the configurations of adding 1 and 5 blocks. [27] suggests that adding 1 block on the $res4$ is slightly better than the others. So our experiments of adding 1 block all target the $res4$ of ResNet. The experiments of adding 5 blocks, on the other hand, are configured by inserting 2 blocks on the $res3$, and 3 blocks on the $res4$, to every other residual block in ResNet-50 and ResNet-101.

**Training:** We use the models pretrained on ImageNet [20] to initialize the weights. The frames of a video are extracted in a *dense* manner. Following [27], we generate 32-frames input clips for models, first randomly crop out 64 consecutive frames from the full-length video and then drop every other frame. The way to choose these 32-frames input clips can be viewed as a temporal augmentation. The crop size for each clip is distributed evenly between 0.08 and 1.25 of the original image and its aspect ratio is chosen randomly between 3/4 and 4/3. Finally we resize it to 224. We use a weight decay of 0.0001 and momentum of 0.9 in default. The strategy of gradual warmup is used in the first ten epochs. The dropout [23] with ratio 0.5 is inserted between average pooling layer and last fully-connected layer. To keep same with [27], we use zero to initialize the weight and bias of the BatchNorm (BN) layer in both CGNL and NL blocks [6]. To train the networks on CUB dataset, we follow the same training strategy above but the final crop size of 448.

**Inference:** The models are tested immediately after training is finished. In [27], spatially fully-convolutional inference [2] is used for NL networks. For these video clips, the shorter side is resized to 256 pixels and use 3 crops to cover the entire spatial size along the longer side. The final prediction is the averaged softmax scores of all clips. For fine-grined classification, we do 1 center-crop testing in size of 448.

### 4.3 Ablation Experiments

**Kernel Functions:** We use three popular kernel functions, namely dot production, embedded Gaussian and Gaussian RBF, in our ablation studies. For dot production, Eq. 12 will be held for direct computation. For embedded Gaussian, the $\alpha_p^2$ will be $\frac{1}{p!}$ in Eq. 9. And for Gaussian RBF, the corresponding formula is defined as Eq. 10. We expend the Taylor series with third order and the hyperparameter $\gamma$ for RBF is set by $1e$-4 [4]. Table 2a suggests that dot production is the best kernel functions for CGNL networks. Such experimental observations are consistent with [27]. The other kernel functions we used, Embedded Gaussion and Gaussian RBF, has a little improvements for performance. Therefore, we choose the dot production as our main experimental configuration for other tasks.

**Grouping:** The grouping strategy is another important technique. On Mini-Kinetics, Table 2d shows that grouping can bring higher accuracy. The improvements brought in by adding groups are larger than those by reducing the channel reduction ratio. The best top1 accuracy is achieved by splitting into 8 groups for CGNL networks. On the other hand, however, it is worthwhile to see if more groups can always improve the results, and Table 2c gives the answer that more groups will hamper the performance improvements. This is actually expected, as the affinity in CGNL block considers the points across channels. When we split the channels into a few groups, it can facilitate the restricted optimization and ease the training. However, if too many groups are adopted, it hinder the affinity to capture the rich correlations between elements across the channels.

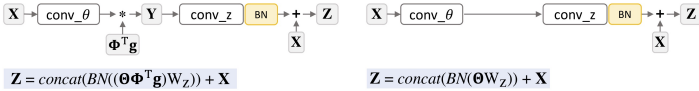

**Figure 3:** The workflow of our CGNL block. The corresponding formula is shown below in a blue tinted box.

**Figure 4:** The workflow of the simple residual block for comparison. The corresponding formula is shown below in a blue tinted box.

**Table 3:** Results comparison of the CGNL block to the simple residual block on CUB dataset.

| model | top1 | top5 |
|---|---|---|
| R-50 | 84.05 | 96.00 |
| + 1 Residual Block | 84.11 | 96.23 |
| + 1 CGNL block | 85.14 | 96.88 |

**Comparison of CGNL Block to Simple Residual Block:** There is a confusion about the efficiency caused by the possibility that the scalars from $\Phi^\top g$ in Eq. 12 could be wiped out by the BN layer. Because according to Algorithm 1 in [11], the output of input $\Theta$ weighted by the scalars $s = \Phi^\top g$ can be approximated to $O = \frac{s\Theta - E(s\Theta)}{\sqrt{Var(s\Theta)}} * \gamma + \beta = \frac{s\Theta - sE(\Theta)}{\sqrt{s^2 Var(\Theta)}} * \gamma + \beta = \frac{\Theta - E(\Theta)}{\sqrt{Var(\Theta)}} * \gamma + \beta$. At first glance, the scalars $s$ is totally erased by BN in this mathmatical process. However, the *de facto* operation of a convolutional module has a process order to aggregate the features. Before passing into the BN layer, the scalars $s$ has already saturated in the input features $\Theta$ and then been transformed into a different feature space by a learnable parameter $\mathbf{W}_z$. In other words, it is $\mathbf{W}_z$ that "protects" $s$ from being erased by BN via the convolutional operation. To eliminate this confusion, we further compare adding 1 CGNL block (with the kernel of dot production) in Fig 3 and adding 1 simple residual block in Fig 4 on CUB dataset in Table 3. The top1 accuracy $84.11\%$ of adding a simple residual block is slightly better than $84.05\%$ of the baseline, but still worse than $85.14\%$ of adding a linear kerenlized CGNL module. We think that the marginal improvement ($84.06\% \rightarrow 84.11\%$) is due to the more parameters from the added simple residual block.

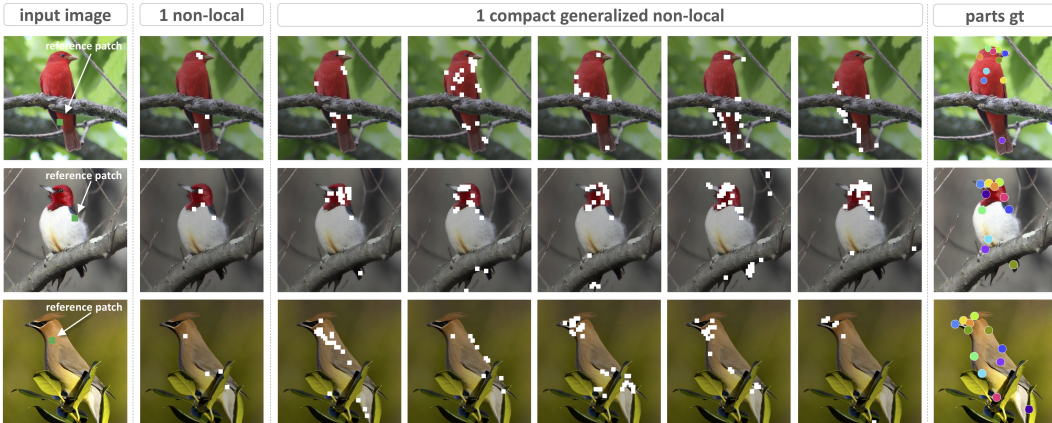

**Figure 5:** Result analysis of the NL block and our CGNL block on CUB. Column 1: the input images with a small *reference patch* (green rectangle), which is used to find the highly related patches (white rectangle). Column 2: the highly related clues for prediction in each feature map found by the NL network. The dimension of self-attention space in NL block is $N \times N$, where $N = HW$. So its visualization only has one column. Columns 3 to 7: the most related patches computed by our compact generalized non-local module. We first pick a reference position in the space of $g$, then we use the corresponding vectors in $\Theta$ and $\Phi$ to compute the attention maps with a threshold (here we use 0.7). Last column: the ground truth of body parts. The highly related areas of CGNL network can easily cover all of the standard parts that provide the prediction clues.

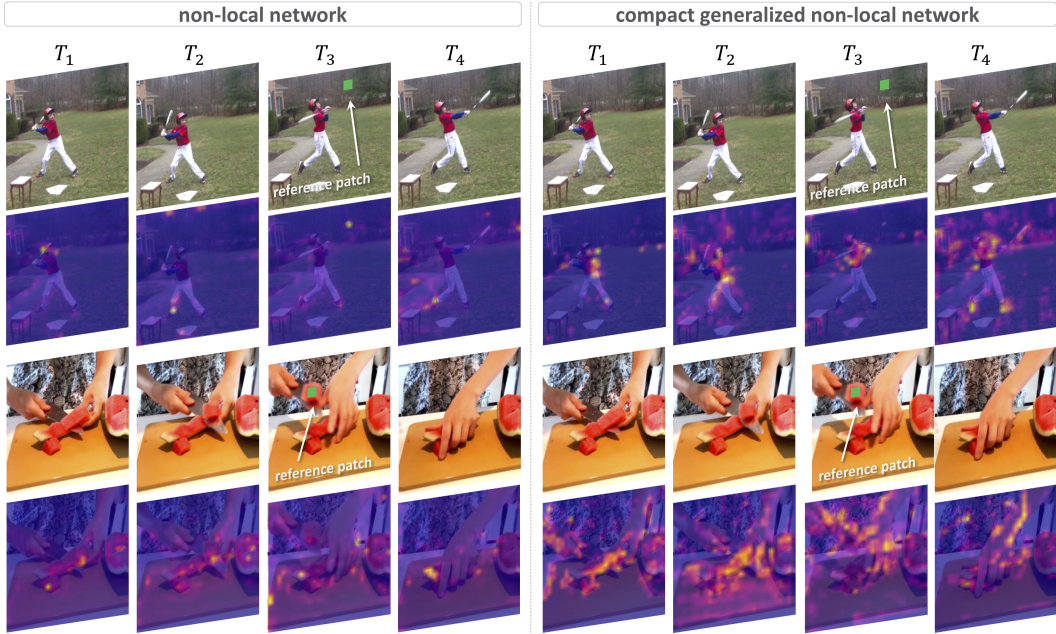

**Figure 6:** Visualization with feature heatmaps. We select a reference patch (*green rectangle*) in one frame, then visualize the high related ares by heatmaps. The CGNL network enjoys capturing dense relationships in feature space than NL networks.

**Table 4: Main results.** Top1 and top5 accuracy (%) on various datasets.

**(a)** Main validation results on Mini-Kinetics. The CGNL networks is build within 8 groups.

| model | top1 | top5 |
|---|---|---|
| R-50 | 75.54 | 92.16 |
| + 1 NL block | 76.53 | 92.90 |
| + 1 CGNL block | 77.76 | 93.18 |
| + 5 NL block | 77.53 | 94.00 |
| + 5 CGNL block | 78.79 | 94.37 |
| R-101 | 77.44 | 93.18 |
| + 1 NL block | 78.02 | 93.86 |
| + 1 CGNL block | 79.54 | 93.84 |
| + 5 NL block | 79.21 | 93.21 |
| + 5 CGNL block | 79.88 | 93.37 |

**(b)** Results on CUB. The CGNL networks are set by 8 channel groups.

| model | top1 | top5 |
|---|---|---|
| R-50 | 84.05 | 96.00 |
| + 1 NL block | 84.79 | 96.76 |
| + 1 CGNL block | 85.14 | 96.88 |
| + 5 NL block | 85.10 | 96.18 |
| + 5 CGNL block | 85.68 | 96.69 |

| model | top1 | top5 |
|---|---|---|
| R-101 | 85.05 | 96.70 |
| + 1 NL block | 85.49 | 97.04 |
| + 1 CGNL block | 86.35 | 97.86 |
| + 5 NL block | 86.10 | 96.35 |
| + 5 CGNL block | 86.24 | 97.23 |

**(c)** Results on COCO. 1 NL or 1 CGNL block is added in Mask R-CNN.

| model | $AP^{box}$ | $AP^{box}_{50}$ | $AP^{box}_{75}$ | $AP^{mask}$ | $AP^{mask}_{50}$ | $AP^{mask}_{75}$ |
|---|---|---|---|---|---|---|
| Baseline | 34.47 | 54.87 | 36.58 | 30.44 | 51.55 | 31.95 |
| + 1 NL block | 35.02 | 55.79 | 37.54 | 30.23 | 52.40 | 32.77 |
| + 1 CGNL block | 35.70 | 56.07 | 38.69 | 31.22 | 52.44 | 32.67 |

## 4.4 Main Results

Table 4a shows that although adding 5 NL and CGNL blocks in the baseline networks can both improve the accuracy, the improvement of CGNL network is larger. The same applies to Table 2b and Table 4b. In experiments on UCF101 and CUB dataset, the similar results are also observed that adding 5 CGNL blocks provides the optimal results both for R-50 and R-101.

Table 4a shows the main results on Mini-Kinetics dataset. Compared to the baseline R-50 whose top1 is 75.54%, adding 1 NL block brings improvement by about 1.0%. Similar results can be found in the experiments based on R-101, where adding 1 CGNL provides about more than 2% improvement, which is larger than that of adding 1NL block. Table 2b shows the main results on the UCF101 dataset, where adding 1CGNL block achieves higher accuracy than adding 1NL block. And Table 4b shows the main results on the CUB dataset. To understand the effects brought by CGNL network, we show the visualization analysis as shown in Fig 5 and Fig 6. Additionally, to investigate the capacity and the generalization ability of our CGNL network. We test them on the task of object detection and

instance segmentation. We add 1 NL and 1 CGNL block in the R-50 backbone for Mask-RCNN [7]. Table 4c shows the main results on COCO2017 dataset [13] by adopting our 1 CGNL block in the backbone of Mask-RCNN [7]. It shows that the performance of adding 1 CGNL block is still better than that of adding 1 NL block.

We observe that adding CGNL block can always obtain better results than adding the NL block with the same blocks number. These experiments suggest that considering the correlations between any two positions across the channels can significantly improve the performance than that of original non-local methods.

## 5 Conclusion

We have introduced a simple approximated formulation of compact generalized non-local operation, and have validated it on the task of fine-grained classification and action recognition from RGB images. Our formulation allows for explicit modeling of rich interdependencies between any positions across channels in the feature space. To ease the heavy computation of generalized non-local operation, we propose a compact representation with the simple matrix production by using Taylor expansion for multiple kernel functions. It is easy to implement and requires little additional parameters, making it an attractive alternative to the original non-local block, which only considers the correlations between two positions along the specific channel. Our model produces competitive or state-of-the-art results on various benchmarked datasets.

## Appendix: Experiments on ImageNet

As a general method, the CGNL block is compatible with complementary techniques developed for the image task of fine-grained classification, temporal feature needed task of action recognition and the basic task of object detection.

**Table 5:** Results on ImageNet. Best top1 and top5 accuracy (%).

| model | top1 | top5 |
|---|---|---|
| R-50 | 76.15 | 92.87 |
| + 1 CGNL block | 77.69 | 93.64 |
| + 1 CGNLx block | 77.32 | 93.46 |
| R-152 | 78.31 | 94.06 |
| + 1 CGNL block | 79.53 | 94.59 |
| + 1 CGNLx block | 79.37 | 94.47 |

In this appendix, we further report the results of our spatial CGNL network on the large-scale ImageNet [20] dataset, which has 1.2 million training images and 50000 images for validation in 1000 object categories. The training strategy and configurations of our CGNL networks is kept same as those in Sec 4, only except the crop size here used for input is 224. For a better demonstration of the generality of our CGNL network, we investigate both adding 1 dot production CGNL block and 1 Gaussian RBF CGNL block (identified by CGNLx) in Table 5. We compare these models with two strong baselines, R-50 and R-152. In Table 5, all the best top1 and top5 accuracies are reported under the single center crop testing. The CGNL networks beat the basemodels by larger than 1 point no matter whichever the dot production or Gaussian RBF plays as the kernel function in the CGNL module.

## Footnotes

[1]Bold capital letters denote a matrix $\mathbf{X}$, bold lower-case letters a column vector $\mathbf{x}$. $\mathbf{x}_i$ represents the $i^{th}$ column of the matrix $\mathbf{X}$. $x_{ij}$ denotes the scalar in the $i^{th}$ row and $j^{th}$ column of the matrix $\mathbf{X}$. All non-bold letters represent scalars. $\mathbf{1}_m \in \mathbb{R}^m$ is a vector of ones. $\mathbf{I}_n \in \mathbb{R}^{n \times n}$ is an identity matrix. $\text{vec}(\mathbf{X})$ denotes the vectorization of matrix $\mathbf{X}$. $\mathbf{X} \circ \mathbf{Y}$ and $\mathbf{X} \otimes \mathbf{Y}$ are the Hadamard and Kronecker products of matrices.

[2]https://github.com/facebookresearch/video-nonlocal-net

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
