[Reviews · NeurIPS 2018]

Reviewer 1



This paper proposes a method to capture long-range relations in images and videos. It's done by modeling interactions between any positions of any channels of a set of feature-maps. It's inspired by non-local module (NL) [31]. While NL aggregates all channel information together to encode position dependencies, the proposed method encode position dependencies between any channel. I like the paper flow: it addresses a valid drawback of the non-local module, with a clear visualization in fig. 1 and proposes a generalized non-local (GNL) module to tackle the problem. Then, it counts the limitation of a naive implementation of the proposed method and tries to overcome it by proposing a compact representation to approximate GNL. The paper seems to be technically correct. the formulations are correct as long as I checked them. It's well-written, well-structured, easy to follow and in details. The related work is okay and up to date. The novelty of this paper is sufficient. It addresses the valid problem of NL in capturing fine-grained interactions of objects and actions. The paper proposes a generalized extension of NL where all the interactions between every position of every channel are modeled. As this Generalization is computationally prohibitive, the paper approximates it by Taylor series up to a certain order. I think this paper seems to be a useful contribution to the community. The experiments are conducted on well-known datasets both in image and vision domain. Experiments are comprehensive on three tasks of fine-grained bird classification, action recognition, and object detection and in most of the cases, the proposed method outperforms others. Ablation study is there and informative. it seems experiments are reproducible. minor: missing closed parentheses in table 1. L69-76 some repetitions.

Reviewer 2



This paper proposes a novel network module to exploit global (non-local) correlations in the feature map for improving ConvNets. The authors focus on the weakness of the non-local (NL) module [31] that the correlations across channels are less taken into account, and then formulate the compact generalized non-local (CGNL) module to remedy the issue through summarizing the previous methods of NL and bilinear pooling [14] in a unified manner. The CGNL is evaluated on thorough experiments for action and fine-grained classification tasks, exhibiting promising performance competitive to the state-of-the-arts. Positives: + The paper is well organized and easy to follow. + The generalized formulation (8,9) to unify bilinear pooling and non-local module is theoretically sound. + Good performance. Negatives: - Less discussion on the linear version of CGNL using dot product for f. - Missing fundamental comparison to the simple ResBlock. The authors nicely present the generalized formulation toward CGNL by unifying the two previous works of bilinear pooling and non-local module. Though the kernelized (non-linear) correlation function f is well theoretically motivated, the actual form of f that achieves the better empirical performance is a “linear” form (dot product). In this regard, the reviewer has the following concerns. - Less discussion about the linear form. If the reviewer correctly understands the CGNL formulation, the linear function f of dot product f (line 204) can greatly simplify the CGNL into Y = X * W_theta * tr[(X*W_phi)’ * (X*W_g)] = X * W_theta * tr[(X’X)* W_g* W_phi’] = s * X * W_theta, where s = tr[(X’X) * W_g * W_phi’]= tr[(X’X)* W] is just a scalar and W = W_g*W_phi’. This reformulation would be beneficial from the following viewpoints. > It reduces the parameters from {W_theta, W_phi, W_g} to {W_theta, W}, which facilitates the implementation. > It is closely related to squeeze-and-excitation (SE) module [9]. The above formulation can be regarded as a bilinear extension of SE from “squeeze” viewpoint since it “squeezes” the feature map X into the bilinear form of X’X while SE simply employs an average-pooling. Such discussions as above would help the readers to further understand the methods and to further extend the method. - Missing comparison. Based on the above discussion, one can think that the baseline for the linear CGNL is a simple ResBlock of Z = BatchNorm( X * W_z ) + X, while the linear CGNL is Z = BatchNorm( s * X * W_theta * W_z ) + X = BatchNorm( s * X * W_tz ) + X. The only difference is the scaling factor s that is also build on X. Through batch normalization, such a scaling might be less effective (during the training) and thus by comparing these closely-related methods, the authors have to clarify its effectiveness of CGNL empirically. Due to this concern, the reviewer can not fairly evaluate the impact of the method on classification performance. [After Rebuttal] The reviewer appreciates the authors’ efforts to perform the comparison experiments in such a short rebuttal period. The comparison with the standard ResBlock clarifies the effectiveness of the proposed method as well as helps us to further understand how it works.

Reviewer 3



This work is an extension of non-local (NL) neural network [31]. Inspired by SENet [9], the authors generalize the non-local module and take the correlations between the positions of any two channels into account. The dense correlations bring improvements for fine-grained visual tasks. Strengths: 1. The motivation of the paper is clear. In order to capture the subtle relationships between objects and scenes, they propose dense correlations to extract information from any channels. The writing is good and concise. Weakness: 1. Technical novelty is limited. There are two contributions in the paper, one is generalized NL module, the other is the compact representation. For the first contribution, it is basically the combination of NL module [31] and group idea [2,8,32,33,35]. For the second contribution, it is the application of compact bilinear pooling from CVPR 2017. 2. The performance improvement is not consistent, and usually quite marginal (below 0.5%). From the visualizations in Figure 4, we may argue that the better performance is brought by the dense connections, not the generalized NL module.